

# May arsenic pollution contribute to limiting *Artemia franciscana* invasion in southern Spain?

Marta I. Sánchez[1],[*], Cathleen Petit[2],[*], Mónica Martínez-Haro[3],[4], Mark A. Taggart[5] and Andy J. Green[1]

[1] Wetland Ecology, Doñana Biological Station, CSIC, Seville, Spain
[2] Université Pierre et Marie Curie (Paris VI), Paris, France
[3] Department of Life Sciences, MARE, University of Coimbra, Coimbra, Portugal
[4] Instituto de Investigación en Recursos Cinegéticos (IREC-CSIC-UCLM-JCCM), Ciudad Real, Spain
[5] Environmental Contamination and Ecological Health, Environmental Research Institute, University of the Highlands and Islands, Scotland, UK
[*] These authors contributed equally to this work.

Corresponding author
Marta I. Sánchez,
marta.sanchez@ebd.csic.es

## ABSTRACT

Limited information exists regarding the complex interactions between biological invasions, pollution, and climate change. Most studies indicate that pollution tends to favor invasive species. Here, we provide evidence that arsenic (As) pollution may have a role in limiting the invasion of the exotic brine shrimp *Artemia franciscana*. We tested As toxicity in natural populations of *Artemia parthenogenetica* (a native taxon) and *A. franciscana* from localities in southern Spain with differing degrees of As contamination. Tests were conducted both under current mean temperature conditions (25 °C), and as per a future climate scenario (i.e., an increase in mean temperature of 4 °C). Acute toxicity was estimated on the basis of the median lethal concentration (at 24 h), and chronic toxicity was evaluated by measuring *Artemia* survival and growth under sublethal exposures (after 26 days). At 25 °C, native *A. parthenogenetica* from the highly polluted Odiel and Tinto estuary was much more resistant to acute As stress ($LC_{50}$-24 h, 24.67 mg $L^{-1}$) than *A. franciscana* (15.78 mg $L^{-1}$) and *A. parthenogenetica* from unpolluted sites (12.04 mg $L^{-1}$)–suggesting that local adaptation to polluted conditions may occur. At 29 °C, resistance of *A. parthenogenetica* from Odiel decreased significantly, and there were no statistical differences in sensitivity between the three species/populations, suggesting that climate change may enhance the probability of invasion. Resistance increased with developmental stage from nauplii to adults, and was extremely high in cysts which still hatched at As concentrations of up to 6400 mg $L^{-1}$. Under sublethal chronic exposure *A. franciscana* performed better (survival and growth) than *A. parthenogenetica*, and both species experienced a faster growth when exposed to As, compared with unexposed (control) individuals, probably due to the hormesis. We discuss the ecological implications of our results.

## INTRODUCTION

A major challenge in invasion ecology is to understand the role of environmental stress in the spread of invasive species (*Alpert, Bone & Holzapfel, 2000*). Most studies to date have focused on biotic factors such as natural enemies (*Torchin & Mitchell, 2004*). Less information exists on anthropogenic factors such as pollution and climate change. The study of interactive effects of these stressors is essential in order to understand and predict the response of organisms and entire ecosystems to invasive species under present and future environmental conditions. Experimental approaches that address realistic ecological scenarios are needed. Organisms are usually exposed to relatively low levels of environmental contaminants so that exposure may be chronic and enduring, but they may also be exposed to high-concentrations for short episodes (when pollutant pulses are released into the environment). Therefore, experiments should include both chronic and acute exposure to contaminants since this may change the outcome of competition between native and invasive species. It is also crucial to consider ongoing climate change. Many projections regarding future climate change suggest that global average temperatures may increase by about 4 °C in the present century; hence, studies that take account of temperature increase within this range are needed.

Brine shrimps *Artemia* spp. (Crustacea, Branchiopoda) are keystone organisms in hypersaline coastal and inland systems around the world. Their principal predators are the waterbirds that are typically abundant in these systems (*Sánchez, Green & Castellanos, 2005*; *Sánchez, Green & Castellanos, 2006*; *Varo et al., 2011*). On the Iberian Peninsula, and across the Mediterranean region, the native taxa are the sexual species *A. salina* and a group of clonal populations classified as *A. parthenogenetica*. However, many populations of native *Artemia* in the Mediterranean region (and worldwide) have been replaced in recent years by the highly invasive *A. franciscana*, which is spread mainly through aquaculture (*Amat et al., 2005*; *Muñoz et al., 2014*). Different populations and species of *Artemia* differ in terms of their sensitivity to metals (e.g., cadmium; *Sarabia et al., 2002*; but see *Leis et al., 2014*) and other pollutants (e.g., organophosphate insecticides; *Varó et al., 1998*). This is particularly relevant when native and invasive species compete, since higher resistance would provide an ecological advantage. Variability in pollution resistance may be related to differences in physiology and metabolism among species in relation to mechanisms for metal detoxification (*Sarabia et al., 2002*). However, variability may also be related to specific environmental conditions and the nature of the pollutant mix experienced by different populations (i.e., to local adaptation).

It has been suggested that local adaptation to contaminated conditions by native *Artemia* from Ria de Aveiro may explain the persistence of the only remaining native population in Portugal (*Rodrigues et al., 2012*; *Pinto, Bio & Hontoria, 2013*). However, this hypothesis has never been tested either for *Artemia* or any other biological invasion. Most studies in invasion ecology focus on the mechanisms allowing an invasive species to dominate a native community, and much less attention has been devoted to the study of factors allowing native populations to survive invasions. In this study we test the hypothesis that the native *Artemia* population in the highly contaminated Odiel and Tinto

estuary (Huelva, Spain) persists due to local adaptation to pollution. The Odiel and Tinto estuary is one of the most polluted estuarine systems in Western Europe (*Grande, Borrego & Morales, 1999*). Both rivers, which drain the Iberian Pyritic Belt, have been contaminated by heavy metals and metalloids for over 4,500 years due to mining activities (*Leblanc et al., 2000*). Although there is no longer active mining, massive amounts of mining waste generated over centuries of exploitation remain in-situ and continue to pollute these rivers (*Younger, 1997*). The estuary is also contaminated by discharges from an industrial complex near the city of Huelva (*Grande, Borrego & Morales, 1999*; *Saez et al., 1999*; *Olías et al., 2004*; *Sarmiento et al., 2009*). Among metals/metalloids, inorganic As is one of the most dangerous in the Odiel and Tinto estuary (*Sarmiento et al., 2009*). Other wetlands in Spain which host *Artemia* have much lower levels of pollutants when compared with Odiel. These include the coastal saltpans in Cadiz Bay (where *A. franciscana* has completely replaced native populations), and Cabo de Gata (where a native *A. parthenogenetica* population still persists) in Andalusia. These sites allow us to compare As toxicity in native and invasive *Artemia,* and to relate this with environmental conditions and pollution loads within these habitats (i.e., to consider local adaptation).

The aim here is to investigate the response of native and invasive *Artemia* to pollution (As) and climate change (increase of 4 °C temperature). We performed acute As toxicity tests in native *A. parthenogenetica* from the highly contaminated Odiel saltpans, in *A. franciscana* from the La Tapa saltpans (Puerto de Santa María, Cadiz Bay) and in *A. parthenogenetica* from the Cabo de Gata saltpans (in Almeria) under two temperature conditions (25 and 29 °C). We assessed the sensitivity of different life cycle stages (nauplii, juveniles, adults and cysts) which may vary in their response to toxicants (*Green, Williams & Pascoe, 1986*; *Mohammed, Halfhide & Elias-Samlalsingh, 2009*). An assessment of the ability of cysts to hatch in polluted conditions is of considerable interest, given the ability of birds to disperse viable cysts (*Sánchez et al., 2012*). Finally, we measured mortality and growth rate in *A. parthenogenetica* from Odiel and *A. franciscana* under chronic exposure conditions.

We hypothesize that: 1) *A. parthenogenetica* from Odiel is locally adapted to high pollution and thus will be more resistant to acute As toxicity than native and invasive populations from less polluted areas; 2) acute toxicity to As will depend on developmental stage–with nauplii being the most sensitive and cysts the most resistant; 3) *A. parthenogenetica* from Odiel will perform better (in terms of mortality and growth rate) in comparison to *A. franciscana* under chronic exposure conditions; and 4) an increase in temperature of 4 °C will increase As toxicity in all *Artemia* populations.

## MATERIAL AND METHODS

### Cyst sampling and processing

Brine shrimp cysts of native *Artemia parthenogenetica* from the highly contaminated Odiel estuary (SW Spain, 37°15′29″N, 6°58′25″W), together with *Artemia franciscana* and *A. parthenogenetica* from less contaminated areas (Puerto de Santa María (Cadiz bay, 36°35.799′N, 6°12.597′W) and Cabo de Gata (Almeria, 36°47′N, 2°14′W), respectively; see Fig. 1) were harvested in January 2014 from the shores of several evaporation ponds of

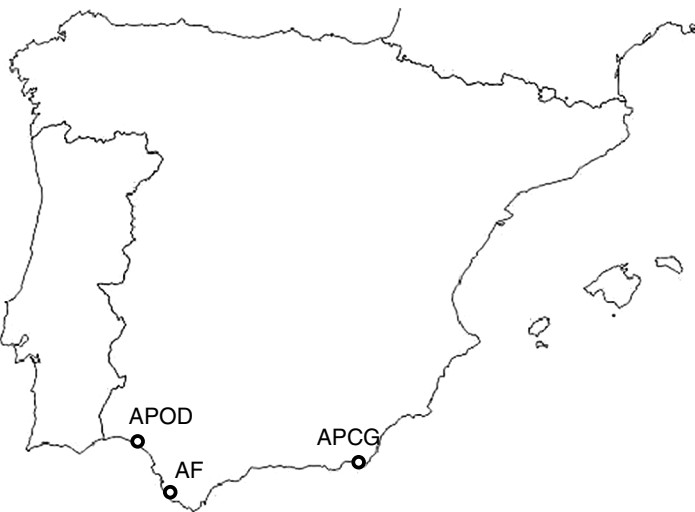

**Figure 1 Study sites.** Locations of the three study populations in southern Spain: APOD (*A. parthenogenetica* from Odiel, Huelva), APCG (*A. parthenogenetica* from Cabo de Gata, Almeria) and AF (*A. franciscana* from Puerto de Santa María, Cádiz).

low-medium salinity (90–150 g L$^{-1}$). The Junta de Andalucía provided permission to sample (1059 Autorización DGGMN). Cysts were transported to the laboratory and sieved through 500, 300, and 100 μm sieves (cyst size is normally ~250 μm). Retained cysts were cleaned by differential flotation in freshwater and saturated brine (after *Sorgeloos et al., 1977*; *Amat, 1985*). Cysts were then dried at 45 °C for 24 h and stored at 5 °C until use in experiments.

### *Artemia* hatching and culture

Cysts were incubated in hatching tanks (Hobby) with artificial sea water (Instant Ocean Sea Salts, 35 g L$^{-1}$) and maintained at 25 and 29 °C for subsequent acute toxicity experiments in climatic chambers. In order to obtain a homogenous population of nauplii instars II and III (as recommended by the standardized ARC-test; Artemia Reference Center, the State University of Ghent in Belgium, *Sorgeloos, Remiche-Van Der Wielen & Persoone (1978)*), hatched larvae were harvested after 48 hours. One portion of the population was used immediately for acute toxicity tests; the other portion was placed in 1 L precipitation tanks at the respective hatching temperature, with a 12:12 photoperiod and gentle aeration for subsequent experiments with juveniles and adults. Salinity was gradually increased (over 3 days) up to 90 g L$^{-1}$. Nauplii were fed with lyophilized green algae *Tetraselmis chuii* (EasyAlgae; Fitoplancton Marino, Cádiz, Spain) solution (algae concentration 0.2 mg mL$^{-1}$). The water was replaced every two days in order to minimize infection by fungus and bacteria.

### Short-term acute toxicity of As (LC$_{50}$-24 h)

Relative mortality of nauplii (median lethal concentration [LC$_{50}$-24 h]) was used to quantify the toxicity to As in the three study populations at two temperatures (25 °C–corresponding with the mean annual temperature in saltpans from south Spain

(*Sánchez, Green & Castellanos, 2006*); and 29 °C–to simulate a 4 °C warming climate change scenario (*IPCC, 2013*)). In addition, $LC_{50}$-24 h was calculated for juvenile *A. parthenogenetica* at 25 °C, juvenile *A. franciscana* at 29 °C and adult *A. franciscana* at 25 °C (these developmental stages/species/temperatures were chosen on the basis of individual availability–certain combinations could not be tested due to high mortality experienced in cultures). Juveniles lack fully developed sexual segments (ovisac or hemipenis).

Arsenic, as reagent-grade sodium arsenate, $NaAsO_2$ (CAS No. 10048-95-0) was used for preparation of a stock solution. The stock solution was kept at ambient temperature and prepared every week for the $LC_{50}$ experiments. Dosing solutions were prepared from the stock solution by mixing different proportions of stock solution and saltwater (Instant Ocean prepared with milliQ) to obtain the desired concentrations based on preliminary tests (Table 1). Final test concentrations were prepared with 35 g $L^{-1}$ salinity for nauplii, and 90 g $L^{-1}$ for juveniles and adults (according to the optimal salinity conditions for the respective stages). Solutions were prepared 24 h prior to use in an experiment, in order to assure an acceptable level of oxygen.

Acute toxicity tests were conducted on nauplii instars II and III in two climatic chambers (at 25 °C and 29 °C) with a 12:12 light:dark cycle. Several ranges of As were used to determine the $LC_{50}$ values for the different populations and temperature conditions (Table 1). Multiwell plates were used for nauplii (2.5 ml) and juveniles (5 ml). For adults, 10 ml sample tubes were used. Nauplii and juveniles were transferred to the multiwell plates with a Pasteur pipette, which carried over less than 0.05 ml of saltwater. For each concentration, including the control (artificial sea water at 35 or 90 g $L^{-1}$ salinity, depending on the developmental stage, without added As), three to six replicates were used (each replicate being composed of 10–12 individuals). After 24 h, the number of dead individuals was recorded. Individuals were considered dead if no movement of the appendages was observed within 10 seconds. For all experiments, preliminary tests were performed in order to adjust the concentrations of As for the final $LC_{50}$ calculations. Immediately before and after the experiment, the oxygen concentration of the different test solutions, including the control, were measured and no substantial variation was observed (mean ± SE: 1.6 ± 0.001 mg $L^{-1}$, 1.7 ± 0.01; respectively). Mortality in control groups without As was no greater than 10 %, if present.

In addition, an acute experiment with cysts of *A. franciscana* and *A. parthenogenetica* from Odiel was performed. Cysts were placed in 0.05 L of artificial saltwater (35 g $L^{-1}$) and stored in climatic chambers at 25 °C or 29 °C. After 5 h, hydrated cysts were selected for the test. Arsenic concentrations ranged from 0 to 6400 mg $L^{-1}$ (4 replicates per concentration, with 30 ± 5 cysts per replicate). Cysts were placed in multiwell plates filled with 5 ml As solution. After three days, the number of hatched nauplii was recorded.

## Chronic sublethal toxicity of As

This experiment was conducted to explore the response of native (*A. parthenogenetica* from the Odiel estuary) and invasive *Artemia* to long-term sublethal exposure to As. The concentration of 0.3 mg $L^{-1}$ As was selected as a compromise based on preliminary
**Table 1 Arsenic concentrations (mg L$^{-1}$) used in LC$_{50}$ tests.**

| Nauplii | | | | | | Juvenile | | Adults |
| --- | --- | --- | --- | --- | --- | --- | --- | --- |
| **25** | | | **29** | | | **25** | **29** | **25** |
| APOD | APCG | AF | APOD | APCG | AF | APOD | AF | AF |
| 0.00 | 0.00 | 0.00 | 0.00 | 0.00 | 0 | 0 | 0.00 | 0.00 |
| 7.20 | 0.47 | 0.12 | 0.47 | 0.47 | 0.47 | 3.75 | 1.88 | 4.69 |
| 14.40 | 0.94 | 0.23 | 0.94 | 0.94 | 0.94 | 7.5 | 3.75 | 9.38 |
| 21.60 | 1.88 | 0.47 | 1.88 | 1.88 | 1.88 | 15 | 7.50 | 18.75 |
| 28.80 | 3.75 | 0.94 | 3.75 | 3.75 | 3.75 | 30 | 15.00 | 37.50 |
| 36.00 | 7.50 | 1.88 | 7.50 | 7.50 | 7.50 | 60 | 30.00 | 75.00 |
| 43.20 | 15.00 | 3.75 | 15.00 | 15.00 | 15.00 | 120 | 60.00 | 150.00 |
| 50.40 | 30.00 | 7.50 | 30.00 | 30.00 | 30.00 | | 120.00 | 300.00 |
| 57.60 | | 15.00 | 60.00 | | | | | 600.00 |
| 64.80 | | 30.00 | 120.00 | | | | | |
| 72.00 | | | | | | | | |

Notes:
 APOD, *A. parthenogenetica* from Odiel; APCG, *A. parthenogenetica* from Cabo de Gata; AF, *A. franciscana*.

analysis of water from Odiel (0.14 ± 0.16 mg L$^{-1}$, n = 4) and LC$_{50}$ tests (7.2 mg L$^{-1}$ was the lowest tested As concentration in which mortality (17.02%) was observed). Fully hatched nauplii of both species were placed at 25 and 29 °C (12:12 photoperiod)–half of them exposed to As and half not (as a control). Salinity was gradually increased from 35 to 90 g L$^{-1}$ over the 10 day period after hatching; 48 specimens per species/ temperature/treatment (control vs As) were randomly selected and individualised for a 25 day experiment. Food (lyophilized *Tetraselmis chuii* at a concentration of 0.20 g L$^{-1}$) was provided every 2 days at the same time that water was changed. Size and mortality were registered every 5 days.

## Statistical analysis

The median acute lethal concentration (LC$_{50}$) and its 95% confidence limits were calculated and compared between different populations, temperatures and developmental stages, using Trimmed Spearman-Karber (TSK) analysis for lethal tests (*Hamilton, Russo & Thurston, 1977*). The criterion of "non-overlapping 95% confidence limits" (CL) was used to determine significant differences between LC$_{50}$ values (lethal concentration necessary to cause 50% mortality; *APHA, 1995*). General Linear Models (GLM; separate slopes) with normal distribution and log-link functions were used to compare slopes of the regression lines of % mortality of different Artemia populations at different As concentrations. Separate slope designs should be used when categorical and continuous predictors (here *Artemia* populations and As concentration, respectively) interact, influencing the responses on the outcome of the model (*Statsoft, 2001*). Two analysis were conducted, one at 25 °C and another at 29 °C, followed by a Bonferroni multiple comparison to test the differences between the slopes of the regression lines. % hatching in the acute experiment was analysed with a GLM, with temperature (25 or 29 °C), population (AP from Odiel and AF) and As concentration (7–11 levels) as categorical

variables, using a normal error distribution and identity link. Two way interactions between the three categorical variables were also included in the model. Residuals were normally distributed. We included partial eta-squared values ($\eta^2$) as a measure of effect sizes. For the chronic toxicity experiment, Cox regression models were used to examine the relationship between temperature, treatment (As exposure, control) and population (AP, AF) as predictors, and survival (time to death from the start of the experiment). Repeated measures ANOVA was performed on growth rate, with temperature, treatment and population as predictor variables. A GLM was also used to analyze the effect of the former variables on the final size (body length, mm) of *Artemia* in the chronic toxicity test, using a normal error distribution and an identity link. Here we also included partial eta-squared values ($\eta^2$).

Statistica 12 software for Windows was used for all statistical analyses.

## RESULTS

### Short-term acute toxicity of As (LC$_{50}$)

The LC$_{50}$ results for the three different *Artemia* populations at two temperatures are shown in Fig. 2. LC$_{50}$ at 25 °C was highest for *A. parthenogenetica* from the polluted area Odiel (24.67 mg L$^{-1}$) followed by *A. franciscana* (15.78 mg L$^{-1}$) and *A. parthenogenetica* from the unpolluted area Cabo de Gata (12.04 mg L$^{-1}$). Based on the criterion of "non-overlapping 95% confidence limits" (*APHA, 1995*) there were strong statistically significant differences in As toxicity between *A. parthenogenetica* from the polluted area and the two other populations. Differences between *A. parthenogenetica* from the uncontaminated area and *A. franciscana* were not so strong but still significant (Fig. 2). The effect of a 4 °C temperature increase on As toxicity differed between populations. While the LC$_{50}$ for *A. parthenogenetica* from Odiel decreased significantly at 29 °C, the LC$_{50}$ for *A. parthenogenetica* from Cabo de Gata increased slightly, but did not change in the case of *A. franciscana*. Overall, differences between populations at 29 °C were not statistically significant (Fig. 2).

The % mortality at different As concentrations at 25 °C (Fig. 3A) didn't differ among populations (GLM: separate-slope model: df = 2, F = 0.728, P = 0.494). However, the interaction between populations and As concentrations was strongly significant (df = 3, F = 29.066, P = 0.0000). The analysis at 29 °C (Fig. 3B) showed statistically significant differences among populations (df: 2, F = 3.725, P = 0.043) and the interaction between *Artemia* population and As concentration was also highly significant (df: 3, F = 40.185, P = 0.0000). % mortality was significantly lower in *A. parthenogenetica* from Odiel compared with *A. parthenogenetica* from Cabo de Gata (P = 0.0216), and higher in *A. franciscana* compared with *A. parthenogenetica* from Cabo de Gata (P = 0.009).

On the other hand, the LC$_{50}$ for *A. parthenogenetica* from Odiel and for *A. franciscana* increased with developmental stage (Fig. 4). LC$_{50}$ was significantly higher in juveniles compared with nauplii for *A. parthenogenetica* at 25 °C (Fig. 4A) and *A. franciscana* at 29 °C (Fig. 4B), as well as in adults compared with nauplii for *A. franciscana* at 25 °C (Fig. 4C). However, the salinity used for nauplii was lower than that for juveniles and adults (see discussion).

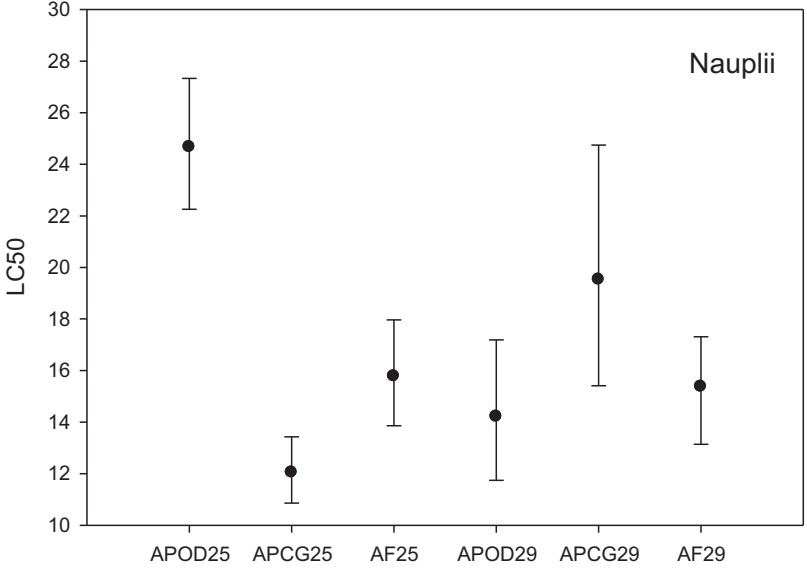

**Figure 2 Median lethal concentration of the three study populations.** $LC_{50}$ values (mg $L^{-1}$) and confidence intervals for nauplii of *A. parthenogenetica* from the contaminated Odiel site (APOD), *A. parthenogenetica* from uncontaminated Cabo de Gata (APCG) and *A. franciscana* (AF) at 25 and 29 °C.

## Acute toxicity test on cysts

Temperature and *Artemia* population had a significant effect on hatching success, with more hatching at 25 °C and higher hatching success (% hatching) for *A. franciscana* compared with *A. parthenogenetica* (Table 2). However, As concentration had no effect, with hatching occurring even at the highest As concentration of 6400 mg $L^{-1}$.

## Long-term toxicity of As

### Survival

The cumulative survival of *A. parthenogenetica* from Odiel and of *A. franciscana* are shown in Figs. 5A and 5B, respectively. According to a Cox regression, there were significant effects due to species, treatment and temperature (Table 3). Survival was higher in *A. franciscana* compared with *A. parthenogenetica*, and the experimental temperature of 29 °C was associated with higher survival compared to 25 °C (Fig. 5). Arsenic exposure significantly reduced survival in both *Artemia* species, compared with controls (Table 3).

### Growth rate

The results of a repeated-measures ANOVA showed significant effects of *Artemia* species, As treatment and temperature on growth rate (Table 4). Individuals grew faster when exposed to As than in controls. In order to exclude the possibility that higher growth rate in exposed individuals was due to selection of bigger, more resistant specimens, we compared the size at the first measurement (21 May) between individuals that were subsequently found dead two days later (23 May) and those that were still alive on that date. There was no significant difference between groups (t = −1.123 with

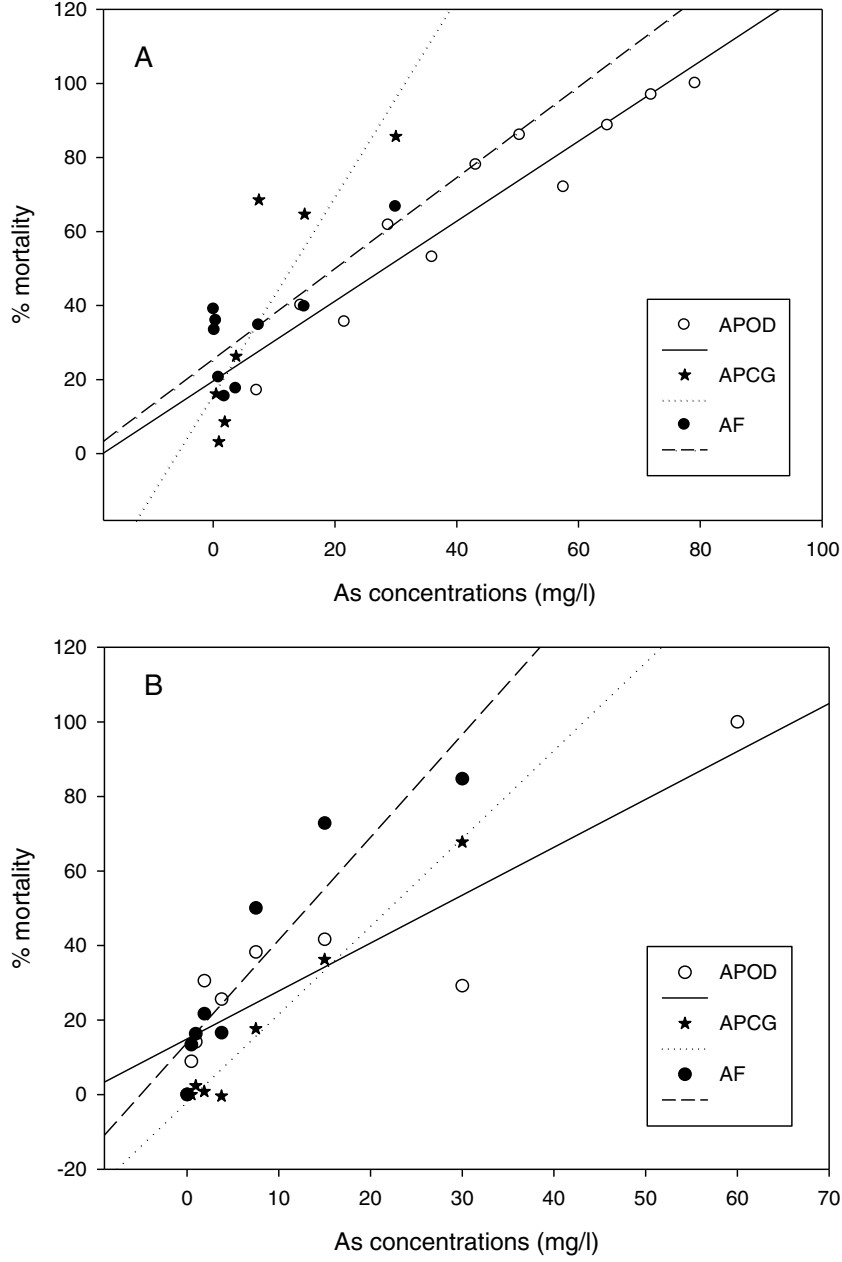

**Figure 3 Linear regression of mortality at different arsenic concentrations.** Linear regression of % mortality of nauplii at 25 °C under different As concentrations for native *A. parthenogenetica* from Odiel (APOD), *A. parthenogenetica* from Cabo de Gata (APCG) and *A. franciscana* (AF). (A) 25 °C; (B) 29 °C.

139 degrees of freedom, P = 0.263). Growth rate was also higher in *A. franciscana* than in *A. parthenogenetica* (Table 4).

### Final size

Results of GLM analyses on final size are shown in Table 5. There were significant effects due to *Artemia* species (*A. franciscana* = 7.07 ± 0.10 mm, *A. parthenogenetica* = 6.89 ± 0.01 mm, mean ± SE), temperature (25 °C = 6.28 ± 0.14 mm, 29 °C = 7.27 ± 0.07 mm) and treatment (control = 6.78 ± 0.09 mm, As treatment = 7.27 ± 0.10 mm).

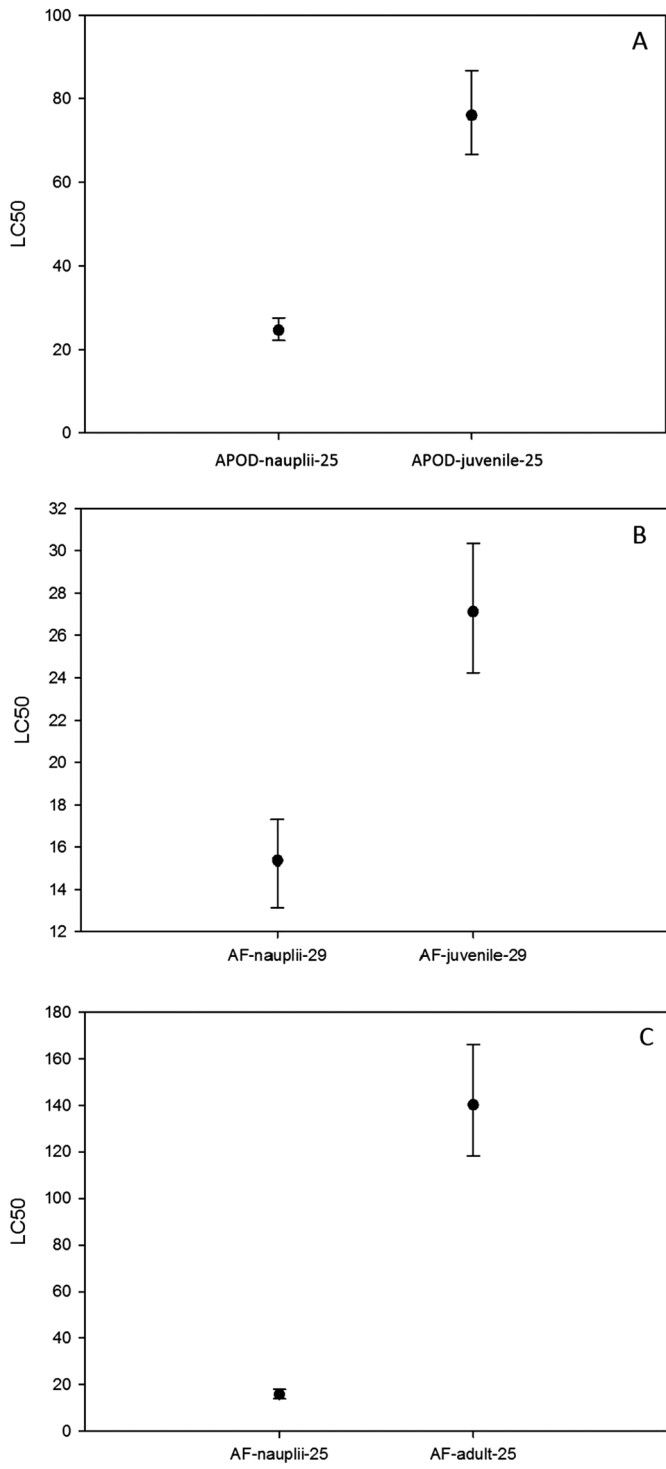

**Figure 4 Median lethal concentration of different developmental stages.** $LC_{50}$ values (mg $L^{-1}$) and confidence intervals of different developmental stages at different temperatures for *A. parthenogenetica* from Odiel (APOD) and *A. franciscana* (AF). Some of the combinations are lacking due to the high mortality experienced in culture.

**Table 2 Generalized Linear Model of hatching cysts.** Results of GLM analysis applied to the % of hatching cysts in relation to temperature (25 °C, 29 °C), population (APOD = *A. parthenogenetica* from Odiel; AF = *A. franciscana*) and As concentration (mg L$^{-1}$). Coefficients for AF and 29 °C are not included because they would be redundant (i.e. they are aliased), but they are effectively zero. The GLM used a normal error distribution and an identity link. Partial eta-squared values ($\eta^2$) are presented as a measure of effect sizes.

| Effects | Level of effect | Estimate | SE | df | F | p | $\eta^2$ |
|---|---|---|---|---|---|---|---|
| Intercept | | 57.8003 | 1.341068 | 1 | 1857.627 | 0.000000 | 0.960666 |
| Temperature | 25 °C | 5.4380 | 1.341068 | 1 | 16.443 | 0.000082 | 0.086210 |
| Population | APOD | −12.4009 | 1.339828 | 1 | 85.667 | 0.000000 | 0.427760 |
| Temperature*concentration | | −0.0019 | 0.000558 | 1 | 11.004 | 0.001149 | 0.170840 |
| Pop*concentration | | 0.0017 | 0.000558 | 1 | 9.569 | 0.002374 | 0.158384 |
| Concentration | | −0.0009 | 0.000558 | 1 | 2.529 | 0.113915 | 0.135572 |
| Temperature*pop | | 4.4175 | 1.110254 | 1 | 15.831 | 0.000109 | 0.126399 |

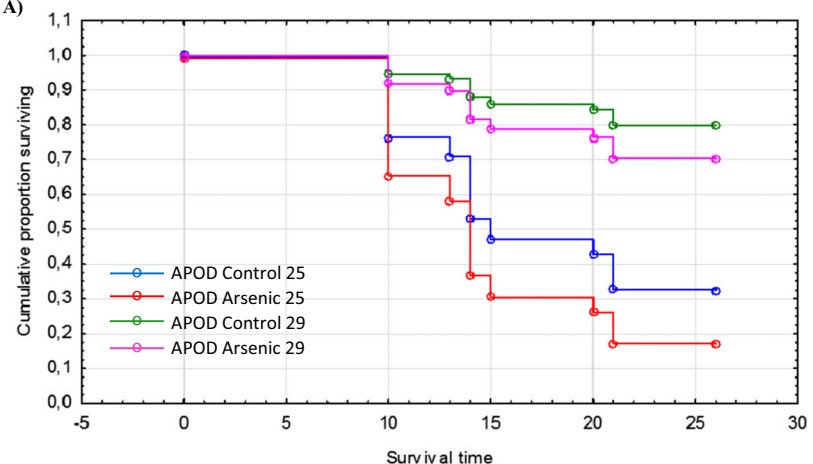

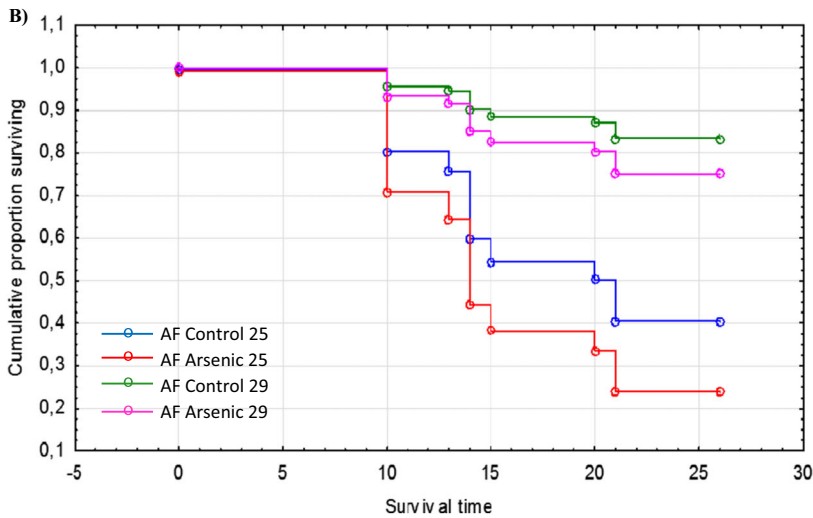

**Figure 5 Survival of native and invasive *Artemia* exposed to arsenic at different temperatures.** Cumulative survival of *A. parthenogenetica* from Odiel (A) and *A. franciscana* (B) exposed to arsenic and in control conditions at 25 and 29 °C. APOD: *A. parthenogenetica* from Odiel; AF: *A. franciscana*.

**Table 3 *Artemia* mortality during long term exposure to As.** Results of Cox proportional hazard regression analysis on *Artemia* survival after As exposure. Based on different temperatures (25 °C, 29 °C), populations (APOD = *A. parthenogenetica* from Odiel; AF = *A. franciscana*) and As concentration (mg L$^{-1}$). Coefficients for AP, As and 29 °C are not included because they would be redundant (i.e. they are aliased), but they are effectively zero.

| Effects | Level of effect | Estimate | SE | Chi-square | P |
|---|---|---|---|---|---|
| Species | AF | −0.105321 | 0.052711 | 3.9924 | 0.045707 |
| Treatment | CONTROL | −0.228578 | 0.053472 | 18.2735 | 0.000019 |
| Temperature | 25 | 0.803722 | 0.066298 | 146.9617 | 0.000000 |

**Note:**
AF, *A. franciscana.*

**Table 4 Growth rate under long term exposure to As.** Results of repeated-measures ANOVA on growth rate for the long term toxicity test, based on different temperatures (25 °C, 29 °C), populations (APOD = *A. parthenogenetica* from Odiel; AF = *A. franciscana*) and As concentration (mg L$^{-1}$). Coefficients for AP, As and 29 °C are not included because they would be redundant (i.e. they are aliased), but they are effectively zero. The GLM used a normal error distribution and an identity link.

| Effect | Level of effect | Parameters | SE | F | p |
|---|---|---|---|---|---|
| Intercept | | 4.209760 | 0.068020 | 3552.689 | 0.000000 |
| Species | AF | 0.340956 | 0.068020 | 8.418 | 0.000022 |
| Treatment | CONTROL | −0.284280 | 0.068020 | 10.232 | 0.000002 |
| Temperature | 25 | 0.481823 | 0.068020 | 47.399 | 0.000000 |
| Species*treatment | 1 | −0.161147 | 0.068020 | 9.216 | 0.000008 |
| Species*temperature | 1 | 0.106952 | 0.068020 | 5.454 | 0.001172 |
| Treatment*temperature | 1 | −0.214388 | 0.068020 | 7.940 | 0.000042 |
| Species*treatment*temperature | 1 | −0.202029 | 0.068020 | 9.942 | 0.000003 |

**Table 5 Results of GLM on final size of *Artemia* after long term exposure to As.** Results of GLM on final size of *Artemia* under different temperatures (25 °C, 29 °C), populations (APOD = *A. parthenogenetica* from Odiel; AF = *A. franciscana*) and As concentration (0.3 mg L$^{-1}$) or control (0 mg L$^{-1}$). Coefficients for APOD, As and 29 °C are not included because they would be redundant (i.e. they are aliased), but they are effectively zero. The GLM used a normal error distribution and a identity link. Partial eta-squared values ($\eta^2$) are presented as a measure of effect sizes.

| Effect | Level of effect | Estimates | SE | df | F | p | $\eta^2$ |
|---|---|---|---|---|---|---|---|
| Intercept | | 6.817502 | 0.074509 | 1 | 8372.187 | 0.000000 | 0.965629 |
| Species | AF | 0.141674 | 0.066386 | 1 | 4.554 | 0.033653 | 0.015053 |
| Treatment | CONTROL | −0.197046 | 0.067657 | 1 | 8.482 | 0.003858 | 0.027676 |
| Temperature | 25 | −0.486999 | 0.073456 | 1 | 43.946 | 0.000000 | 0.128516 |
| Error | | | | 298 | | | |

## DISCUSSION

### Response of *Artemia* to acute As stress and its ecological implications

Disturbance has been widely recognized as a major determinant in the establishment of non-indigenous species (*D'Antonio, 2000*; *Piola & Johnston, 2008*). Pollution, in particular with heavy metals, is one of the most important anthropogenic disturbances in coastal

ecosystems worldwide (*Scanes, 1996*; *Hall, Scott & Killen, 1998*; *Sarmiento et al., 2009*). However, the specific role of pollution in facilitating or preventing invasion has been largely overlooked. Results of the few studies that have addressed how invasive species respond to pollution conclude that it favors them. Most of these studies have been conducted with copper, but little information exists on other metals or metalloids, such as arsenic. For example, *Piola & Johnston (2006)* showed that non-indigenous species have a greater tolerance to copper pollution when compared to closely related native species. *Piola & Johnston (2008)* studied the effect of heavy metal pollution on the diversity of marine hard-substrate assemblages and showed that increasing pollution exposure decreased native species diversity by between 33% and 50% while no effect was detected for non-indigenous species, suggesting that the latter are more tolerant to metal pollution relative to their native counterparts. Similar results were found by *Crooks, Chang & Ruiz (2011)* studying marine invertebrates in San Francisco Bay; they found that copper exposure significantly decreased native species richness but didn't affect exotic species richness. However, information is still scarce and probably biased because most studies focus on successful invasions and very few on systems which have resisted the establishment of an invasive species (*Rodrigues et al., 2012*).

At 25 °C, we found that native *A. parthenogenetica* from Odiel was more tolerant to As than an invasive *A. franciscana* population and an *A. parthenogenetica* population from a relatively unpolluted area. Our results suggest that *A. parthenogenetica* from Odiel is locally adapted to withstand high pollution levels. In turn, this supports the hypothesis of *Rodrigues et al. (2012)* that some populations of native *Artemia* may persist because they are adapted to pollution that may limit the invasion of *A. franciscana*. *Rodrigues et al. (2012)* referred to high mercury pollution in the Ria de Aveiro in Portugal. This site holds the last known population of *A. parthenogenetica* in that country, wherein all other saltworks have been invaded by *A. franciscana* (*Amat et al., 2005*; *Amat et al., 2007*).

Our study represents the first to test this local adaptation hypothesis in *Artemia* and the first to evaluate the response of *Artemia* to acute As exposure. Most studies in invasion ecology focus on the mechanisms by which an invasive species comes to dominate a native community. Much less attention has been paid to the factors that allow native populations to survive an invasion (*Rodrigues et al., 2012*). The *A. parthenogenetica* population from Odiel is surrounded by sites already invaded by *A. franciscana* (including Isla Cristina 30 km away and Cádiz Bay at 90 km). This suggests that pollution may indeed help explain why the native species has so far persisted.

Migratory waterbirds (such as shorebirds and flamingos) are highly effective at dispersing cysts of both native and invasive *Artemia* between different localities (*Sánchez et al., 2007*; *Muñoz et al., 2014*). The persistence of *A. parthenogenetica* in the much less polluted Cabo de Gata (Almeria) may be related to a dispersal limitation, i.e., it is outside the main shorebird migratory flyway (East Atlantic Flyway, *Green et al., 2005*) and is over 200 km away from the nearest *A. franciscana* population. However, the ongoing survival of remaining native *Artemia* populations in the Mediterranean region is also linked to an absence of aquaculture in surrounding wetlands. This has prevented the large scale introduction of *A. franciscana* as fish food (*Amat et al., 2005*) to sites such as Odiel and Cabo de Gata.

Recent studies have compared the sensitivity of native and invasive species to different contaminants; *Leis et al. (2014)* examined the toxicity of Hg, Cd and Cr to native *A. parthenogenetica* from Italy and to *A. franciscana*, and didn't find differences between populations. *Varó et al. (2015)* showed that *A. franciscana* is less affected (in terms of survival and fecundity) by exposure to the pesticide chlorpyrifos than *A. parthenogenetica* (diploid). However, to fully test the resistance hypothesis it is necessary to compare populations naturally exposed to different degrees of pollution. Here we provide a first approach to the problem. One important difference between our system and previously studied sites is acclimation time–i.e., the period over which species have been exposed to pollution. All previous studies generally consider scenarios involving recent environmental pollution or emerging pollutants (for example *Varó et al., 2015*). Pollution tends to have occurred since the middle of the last century, and in any case, no more than for 200 years (i.e., since the start of the Industrial Revolution). A substantially different scenario may exist when native communities have been exposed to pollutants for millennia (i.e., in areas with prehistoric mining activities)–as is the case in the Odiel and Tinto river basins (*Nocete, 2006*). Under these circumstances, we may expect native communities to be highly adapted, and therefore more resistant to the establishment of newly arriving non indigenous species.

## Sensitivity of different *Artemia* developmental stages to As

We found that cysts of both *A. parthenogenetica* and *A. franciscana* were extremely resistant to As. This is likely due to the highly impermeable chorion that acts as a barrier against toxicants (as demonstrated by *Varó et al. (2006)* with organophosphate pesticides). Similarly, *Sarabia et al. (2003)* found no effect of Cd on hatching success of *Artemia*. However, other authors have reported a strong effect regarding Cd, Zn and Cu (*Bagshaw et al., 1986*; *MacRae & Pandey, 1991*; *Rafiee et al., 1986*). It has been suggested that differences in hatching success of cysts may also be related to differences in cyst structure, metabolism and physiology among species (*Varó et al., 2006*). In addition to these factors, the previous environmental conditions (i.e., levels of pollution) experienced by the species/strain could also play a decisive role. On the basis of $LC_{50}$ values here, we found nauplii to be the most sensitive developmental stage, followed by juveniles and adults. Although different salinities were used in experiments for nauplii (35 g/l) and juvenile/adults (100 g/l) based on the optimal salinity conditions of these different developmental stages, this is not expected to affect our results. There are field studies showing a relationship between salinity and arsenic in animal tissues (e.g. *Larsen & Francesconi, 2003*) or environmental samples (e.g. sediments, *Kulp et al., 2007*), but such salinity effects are likely to be because evaporation increases the concentration of dissolved toxic substances as well as non-toxic salts, or because salinity induces changes in the arsenite-oxidizing and arsenate-reducing microbial community. To our knowledge, no evidence exists that salinity *per-se* influences As toxicity, in the absence of confounding factors. Our results may instead be explained by the ratio between gut volume and body mass (*Navarro, Ireland & Tytler, 1993*), since the gut is highly permeable compared with the external

cuticle (*Croghan, 1958*), with ion exchange in nauplii occurring three times faster than in adults (*Thuet, Motais & Maetz, 1968*).

## The effect of an increase in temperature

The sensitivity of *A. parthenogenetica* from Odiel to As increased significantly as we moved from 25 to 29 °C. The lower temperature currently represents the mean temperature in the field for the Odiel population (*Varo et al., 2011*). Temperature increases have often been found to increase contaminant toxicity (*Cairns, Heath & Parker, 1975*; *Bat et al., 2000*), and to decrease dissolved oxygen concentration, especially at high salinities. Temperature increases within typical ranges in biological systems may have little effect on metal speciation (*Bervoets & Blust, 1999*; *Hassler, Slaveykova & Wilkinson, 2004*), but may influence toxicity through physiological mechanisms. This is particularly true in ectotherms (*Sokolova & Lannig, 2008* for review), since their body temperature depends on that of the environment. Hence, changes in external temperature cause changes in metabolic rates (*Hochachka & Somero, 2002*) and consequently metal uptake (*Sokolova & Lannig, 2008* and references therein). The permeability of diffusion membranes in *Artemia* spp. is also known to increase with temperature (*Navarro, Ireland & Tytler, 1993*).

The response to temperature of *A. parthenogenetica* from Cabo de Gata was substantially different–with a slight (but significant) decrease in As sensitivity at higher temperatures. This is not the first study to find that different populations of the same *Artemia* species respond differently to abiotic conditions such as temperature (*Browne & Wanigasekera, 2000*). Decreasing toxicity with increasing temperature has also been described for some organic pollutants such as DDT (*Cairns, Heath & Parker, 1975*), but it is not common in metals/metalloids. *A. parthenogenetica* from Cabo de Gata may have higher thermo-tolerance than *A. parthenogenetica* from Odiel. It has been shown that among parthenogenetic strains, polyploids are better suited to temperature extremes (both high and low) than diploids (*Zhang & Lefcort, 1991*); however, ploidy cannot explain the observed differences because the Cabo de Gata population is 2 n whilst the Odiel population is 2 n plus a small fraction of 4 n (*Amat et al., 2005*). The differences between *A. parthenogenetica* from Odiel and Cabo de Gata may perhaps be related to a trade-off between pollution resistance and the ability to cope with another environmental stressor (temperature) in the Odiel population. Pollution resistance has been found to trade-off against fitness traits such as growth and fecundity in many different organisms (see below). Toxicity was not affected by temperature in the case of *A. franciscana*, which is in agreement with the higher temperature tolerance of this invasive species (*Browne & Wanigasekera, 2000*; *Zerebecki & Sorte, 2011*). Such a response, in which pollutants have constant toxicity irrespective of temperature, is rarely found in aquatic ectotherms; however, this was also the case in *Daphnia pulex* exposed to Cu (*Boeckman & Bidwell, 2006*). Overall, our results suggest that global warming may be expected to favour the invasion of *A. franciscana* in highly contaminated areas such as Odiel.

## Long-term sublethal exposure to As

The results of long-term (sublethal) exposure showed that *A. franciscana* performs better (higher survival and growth) than *A. parthenogenetica* under chronic stress. This is not surprising, as several studies using sexual and asexual *Artemia* populations from the Old World show that the competitive ability of *A. franciscana* is higher than that of *A. salina* and parthenogenetic strains (*Browne, 1980*; *Browne & Halanych, 1989*; *Amat et al., 2007*). Similar results demonstrating a low impact due to chronic As stress (0.24 mg $L^{-1}$, equivalent to that of our study) were also obtained by *Brix, Cardwell & Adams (2003)* with *A. franciscana* from the native area in Great Salt Lake (Utah, U.S.A). The introduced *A. franciscana* we studied is closely related to the Great Salt Lake population, owing to the trade in cysts from that lake for aquaculture (*Muñoz et al., 2014*). Thus, resistance to chronic As stress was probably selected for long before their introduction from North America. A similar scenario has been suggested previously for other invasive species–such as highly Cu-resistant introduced populations of the bryozoan *Bugula neritina*, which originate from polluted ports and harbours (*Piola & Johnston, 2006*).

Our results also suggest faster growth in individuals exposed to As than in controls. This may be related to hormesis, i.e., the stimulatory effect caused by low levels of toxic agents (*Stebbing, 1982*). Growth stimulatory responses to low doses of various chemicals were first observed in yeast (*Schulz, 1888* 'Arndt-Schulz law') and this has been demonstrated for a wide range of organisms (including bacteria, protozoan, plants, algae, invertebrates and vertebrates), endpoints (including growth, reproduction, behaviour, survival, physiology), and toxicants (metals, pesticides, effluents, etc.) (reviewed in *Calabrese & Baldwin, 2003*). Arsenate is also a chemical analogue of phosphate (*Tawfik & Viola, 2011*), so at low doses, physiological processes involving phosphate may "inadvertently" utilise arsenate.

The As concentration used in our experiments was similar to that recorded in water from the Odiel site (maximum of 0.23 mg $L^{-1}$) in order to make our results as relevant as possible to real field conditions. However, the bioavailability of this metalloid is expected to be significantly higher in natural conditions for several reasons: 1) The concentration of As in the sediments of the Odiel study area is often high (maximum of 123 mg $L^{-1}$) and *Artemia* are known to feed on detritus (*Sánchez et al., 2013*) which is likely to be polluted. 2) In the experiment we used commercial lyophilized algae which were not a source of As, while in natural conditions *Artemia* feed on phytoplankton which is able to accumulate certain metals. 3) Odiel has extremely high concentrations of phosphates (Sánchez et al., 2011, unpublished data) and phosphorous is known to increase the bioavailability of As (*Bolan et al., 2013*). Further experiments under conditions that better reflect all potential As sources that exist in the field will be important in order to fully assess the potential of *A. franciscana* to invade hypersaline complexes within the Odiel estuary.

## CONCLUSION

This study represents the first step to investigate the "*pollution resistance hypothesis*" (*Rodrigues et al., 2012*) and the effect of acute As exposure in *Artemia*. Moreover,

although several studies have focused on the impact of metals on *Artemia*, very few have compared toxicity between native and invasive species nor considered realistic different scenarios relevant to climate change. We found support for the idea that *A. parthenogenetica* from Odiel is locally adapted to elevated pollution. Our results also suggest that climate change would increase the susceptibility of pollution-resistant *A. parthenogenetica* populations to invasion by *A. franciscana*. This study highlights the importance of simultaneously considering the effect of different stressors so that future risks to organisms and ecosystems can be better understood. It also illustrates the value of focusing on systems that are resisting invasions, and not just those which have already been invaded.

## ACKNOWLEDGEMENTS

E. Martínez, Director of Marismas del Odiel Natural Park, provided permission to work in the salt ponds. Raquel López Luque and staff of the Aquatic Ecology Laboratory of the EBD-CSIC helped with the experiments.

### Funding

This work was funded by the Spanish Ministry of Economy and Competitiveness, through the Severo Ochoa Program for Centres of Excellence in R+D+I (SEV-2012-0262), through the I+D+i National Plan (Project CGL2013-47674-P) and by the 7th Framework Programme (FP7 2007-2013) of the European Commission through a Marie Curie Intra-European Fellowship for Career Development (PIEF-GA-2011-299747). M.I. Sánchez was supported by a Ramón y Cajal postdoctoral contract from the Spanish Ministry of Science and Innovation (MICINN). M. Martínez-Haro was supported by a postdoctoral contract funded by the Junta de Comunidades de Castilla-La Mancha (POST 2014/7780). Currently, MMH holds a Juan de la Cierva research contract (IJCI-2014-20171) awarded by the Spanish Ministry of Science and Innovation-European Social Fund. The funders had no role in study design, data collection and analysis, decision to publish, or preparation of the manuscript.

### Grant Disclosures

The following grant information was disclosed by the authors:
Centres of Excellence in R+D+I: SEV-2012-0262.
I+D+i National Plan: CGL2013-47674-P.
7th Framework Programme: FP7 2007-2013.
Marie Curie Intra-European Fellowship for Career Development: PIEF-GA-2011-299747.
Junta de Comunidades de Castilla-La Mancha: POST 2014/7780.
Juan de la Cierva research contract: IJCI-2014-20171.

### Competing Interests

Marta I. Sánchez is an Academic Editor for PeerJ.

## Author Contributions

- Marta I. Sánchez conceived and designed the experiments, performed the experiments, analyzed the data, wrote the paper, prepared figures and/or tables, reviewed drafts of the paper.
- Cathleen Petit performed the experiments, prepared figures and/or tables, reviewed drafts of the paper.
- Mónica Martínez-Haro conceived and designed the experiments, performed the experiments, reviewed drafts of the paper.
- Mark A. Taggart contributed reagents/materials/analysis tools, reviewed drafts of the paper.
- Andy J. Green conceived and designed the experiments, analyzed the data, wrote the paper, reviewed drafts of the paper.

## Field Study Permissions

The following information was supplied relating to field study approvals (i.e., approving body and any reference numbers):

Consejería de Medio Ambiente y Ordenación del Territorio, Dirección general de Gestión del Medio Natural, Junta de Andalucía: 1059 Autorización DGGMN.

## Data Deposition

The dataset has been supplied as a Supplemental Dataset file.

## Supplemental Information

Supplemental information for this article can be found online at http://dx.doi.org/10.7717/peerj.1703#supplemental-information.

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
