# Peer review of "May arsenic pollution contribute to limiting Artemia franciscana invasion in southern Spain?"

_PeerJ, doi:10.7717/peerj.1703_

## Round 0.1 · original submission · Major Revisions

· Academic Editor

Major Revisions

The reviewers and I appreciate the work you have accomplished. While both reviewers see strengths in your paper, both made substantial criticisms and suggestions for improvement. I therefore invite you to revise your paper in light of the comments made.

Reviewer 1 ·

Basic reporting

The manuscript follow the requirements of the editorial criteria and author instruction of the Peer J journal.
In what concerns English writing I would like the authors to revise the following sentences with were not clear to me and I think to other readers or have small mistakes:
Abstract; Line 31-34 should be rewritten.
Introduction; Line 57-59.The sentence does not sound correctly I suggest: The study of interactive effects of these stressors is essential in order to understand and predict the response…..
Introduction; Line 60. Change to: Experimental approaches that address realistic ecological scenarios are needed.
Introduction; Line 94. I suggest to substitute against by to.
Material and Methods; Line 254. Substitute the “an experimental…” for “the experimental…”
Discussion; Line 277 an enter was done after “such as” and the sentence should continue on the same line.
Discussion, Line 323. The sentence that starts with “One important” should have a space after the full stop.
Figures:
Fig 4 should be revised because in the text the authors mentioned Fig. 4a, b, c and in the figure that it is not clear. The combination of AP 29 (nauplii) and AP 29 (juvenile) were not shown, why? That information should be stated in the Figure caption.

Experimental design

The experimental design was careful planned as well as the working hypotheses. The discussion present and answer correctly the objective of the work described in introduction.
The section of statistical analysis was the one that raised more concern and in my opinion it should be rewritten in order to carefully explain the statistical analysis performed to the data obtained. The author’s don´t mentioned in the statistical analysis section of Material and Methods, the Cox regression performed to evaluate the survival analysis as well the repeat measures ANOVA s. Even the GLM should be better explained what linear model applied and how you have done. Something like: Data was analyzed with GLM and factors (Temperature (25 and 29ºC), Population (A. parthenogenetica and A. franciscana) and As Concentration) were analyzed by the following model: xxxxx (equation) and how you have dealt with the residual distribution.
In Results section line 234-238 the authors mentioned differences between how the two populations of Artemia exposed to As in two temperatures and they respond differently and the authors said that differences were significantly in one cases and not significant in another case. What were the statistical test performed for supporting you statement, if they have already test it before, the authors should remember the audience in order not get it lost, otherwise you should add that statistical test to the statistical analysis section of the Material and Methods.

Validity of the findings

Very interesting research and very interesting results obtained. The climate change effects on biological communities sets the conditions for development of non- native species was very important issue discussed in literature these days. The article was well written with some exceptions mentioned above.

Reviewer 2 ·

Basic reporting

The paper is interesting and addresses the (Artemia) invasiveness from an original point of view, namely that of comparing invasion against locally adapted populations. It is well written and structured.

Experimental design

Some data seem to be of fragmentary nature with experimental design not considering all potential combinations (see for example Fig. 3: mortality test at different As concentrations only performed at 25ºC; or Fig. 4: median lethal concentrations of AP adults not tested, AF adults at 29 ºC not tested, AF juveniles at 25 ºC not tested). Besides, no references to real toxic concentrations, beyond nominal ones, are reported, thus severely limiting the strength of the results. Reference to high mortalities in L164 need also some further attention thereafter in view of apparent inconsistencies in the results. For example: are exposed individuals bigger and faster-growing due to hormesis as proposed, or may this be the result of selection of the biggest more resistant individuals? Envisaging the global warming scenario, another information missing is that of the thermal preferendum of the strains. Parthenogenetic strains (especially tetraploids) have a higher thermo-preferendum than Old World bisexuals, with A. franciscana thriving successfully in all conditions (wider thermo-preferendum). I think it may be worth adding a few sentences on this topic, and also reviewing the results from Fig. 2 taking into account this consideration, although I am afraid that there will be still no explanation for the sensitivity behaviour of both parthenogenetic strains at 29 ºC .

Validity of the findings

All in all, this is not to say that the work is invalid, but authors should perhaps emphasize that evidences presented here are a first approach to the problem rather than a complete categorical corroboration of hypotheses.

Additional comments

Statistical section is incomplete without references to Cox regression and Repeated measures ANOVA. As a suggestion regarding GLM, since most effects are significant throughout: would not it be worth to study the relative weight (potence) of the factors through for example eta square?

Reference made to discussion from the results section (L242) on the different salinity used for the different developmental stages is not followed or referred to thereafter.

It may be worth considering the role of nucal salt gland in osmorregulation of the nauplii when discussing their higher sensitivity.

Reference to the Great Salt Lahe origin of the A. franciscana population (L395) is speculative since A. franciscana from other sites (i.e. San Francisco Bay) is also used for aquaculture purposes.

Use of italics should be reviewed throughout the text, especially in the references section.

Edit the use and consistency of abbreviations in tables and figures.

L230: avoid the term “marginally” significant.

L237: significance is not shown in the figure.

---

## Round 0.2 · accepted · Accept

· Academic Editor

Accept

I feel that you have successfully addressed the concerns of the reviewers and I am happy to accept this revised version now.